# Neural Network, Nonlinear-Fitting, Sliding Mode, Event-Triggered Control under Abnormal Input for Port Artificial Intelligence Transportation Robots

**Yaping Zhu [1], Qiang Zhang [1,2,*], Yang Liu [3], Yancai Hu [1] and Sihang Zhang [1]**

[1] School of Navigation and Shipping, Shandong Jiaotong University, 1508 Hexing Road, Huancui District, Weihai 264209, China

[2] Shandong Intelligent Transportation Key Laboratory of Shandong Jiaotong University, 5001 Haitang Road, Changqing University Science Park, Jinan 250357, China

[3] Department of Maritime Transportation, Mokpo National Maritime University, 91 Haeyangdaehang-ro, Mokpo City 58628, Republic of Korea

**\*** Correspondence: zq20060054@163.com

**Abstract:** A new control algorithm was designed to solve the problems of actuator physical failure, remote network attack, and sudden change in trajectory curvature when a port's artificial intelligence-based transportation robots track transportation in a freight yard. First of all, the nonlinear, redundant, saturated sliding surface was designed based on the redundant information of sliding mode control caused by the finite nature of control performance; the dynamic acceleration characteristic of super-twisted sliding mode reaching law was considered to optimize the control high frequency change caused by trajectory mutation; and an improved super-twist reaching law was designed. Then, a nonlinear factor was designed to construct a nonlinear, fault-tolerant filtering mechanism to compensate for the abnormal part of the unknown input that cannot be executed by adaptive neural network reconstruction. On this basis, the finite-time technology and parameter-event-triggered mechanism were combined to reduce the dependence on communication resources. As a result, the design underwent simulation verification to verify its effectiveness and superiority. In the comparative simulation, under a consistent probability of a network attack, the tracking accuracy of the algorithm proposed in this paper was 22.65%, 12.69% and 11.48% higher those that of the traditional algorithms.

**Keywords:** nonlinear-fitting redundant sliding mode; event-triggered; abnormal input; neural network; artificial intelligence transportation robots; track tracking

## 1. Introduction

With the advent of the era of Industry 4.0, artificial intelligence transportation robots have gradually matured and have become widely used in ports, logistics, and other freight-related situations [1]. Tianjin Port uses cutting-edge technologies, such as unmanned driving, artificial intelligence, and big data, to replace traditional transportation equipment. There are artificial intelligence transportation robots and a high level of automation for terminal operations, and it serves as a reference for the construction and development of domestic container automation terminals. In 2020, Hefei Port introduced artificial intelligence transportation robots, which continuously improve and expand their perception capabilities in a real operating environment through fusion algorithms, and realize the trajectory prediction of surrounding traffic participants. They not only realize safe and stable operation, but also take into account operational efficiency to ensure efficient and smooth operation. Other port yards have gradually introduced artificial intelligence transportation robots, as shown in Figure 1. However, unknown control anomalies caused by dynamic uncertainty, control signal transmission noise, network attacks, and program

faults in the real environment make it impossible for artificial intelligence transportation robots to achieve precise control. In order to ensure the efficient operation of artificial intelligence transportation robots in ports and terminals, high-precision trajectory tracking control is an urgent problem to be solved.

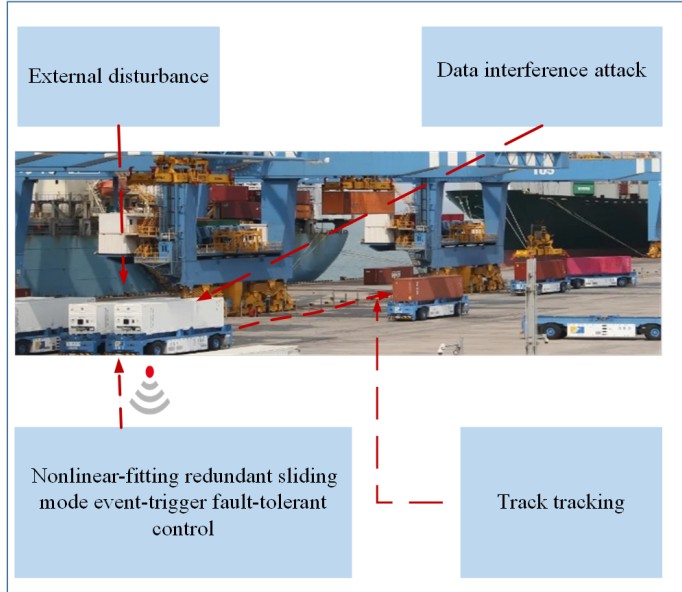

**Figure 1.** Artificial intelligence transportation robots: operational diagram of the Qingdao Port Freight Yard.

Nowadays, the robustness of mature control algorithms such as PID [2] and linear feedback [3] cannot meet the actual needs of jobs in complex scenes. Sliding mode control [4] is recognized by the control community more and more because of its good robustness and simple structure. It is worth noting that how to effectively reduce the chattering effect of the algorithm itself has become a main premise of practical engineering applications. Therefore, under the premise of ensuring robustness, some scholars used the high-dimensional sliding surface [5] to optimize the buffeting threshold, and others used the dynamic characteristics of the sliding mode to design the variable-speed reaching law [6–10] to slow the speed of the near sliding surface to reduce chattering.

It is worth noting that reducing chattering will reduce the robust performance of sliding mode control to a certain extent. The interference caused by dynamic uncertainty especially will mean the robustness of the algorithm cannot be fully brought into play. Therefore, Baek et al. [11] established a stochastic stability judgment mechanism for uncertainty, and used a time-delay estimation scheme combined with adaptive technology to achieve good asymptotic stochastic stability. Niu et al. [12] used neural network approximation to realize on-line robust sliding mode adaptive control. However, adaptive conservatism will waste control resources, and intelligent algorithms require high performance of the controller and need to be further optimized. Zhang et al. [13] proposed a high-order, fast, non-singular, terminal sliding mode controller based on a double-disturbance observer, which effectively weakens the chattering phenomenon of the system. Wang et al. [14] proposed an adaptive, proportional-integral-derivative, fractional-order, non-singular, terminal sliding mode control method based on time-delay estimation, which realizes timely and accurate adjustment of the control gain of the robust term. Shao et al. [15] proposed an adaptive, recursive terminal sliding mode controller. The fast, non-singular end sliding function, and the recursive integral end sliding function were designed by using the recursive structure, so that the sliding surface reaches continuously, which significantly improves the tracking-error-convergence speed and the anti-interference ability speed-wise.

In practical engineering, robot tasks are required to be timely, and most of the above algorithms are asymptotically stable. Therefore, how to complete the tracking movement

within a specified time is an urgent need for the project. Therefore, finite-time technology was proposed and applied in control systems [16–18]. Wang et al. [19] used a variable division technique and fuzzy control. Wang et al. [20] took advantage of the approximation of fuzzy logic systems. Fang et al. [21] gave sufficient conditions for practical fixed-time stability, as did Ba et al. [22], with the help of a neural network and backstepping technology. Zuo [23] carried out finite-time control design for second-order nonlinear systems with uncertainties and disturbances. However, the convergence time of the system with finite-time stability increases with the initial error, which greatly weakens the convergence performance of the system. At the same time, in actual engineering, there is are artificial intelligence transportation robots that have physical limits in the initial stage, and the error is mostly the maximum error of the control task, but this characteristic has not been paid attention. The use of limited control resources to complete the control an abnormal environment has not been further considered.

When transporting in a port's cargo yard, due to the automatic operation of multi-frequency scanning for a long time, the equipment will be worn out, which greatly reduces the control accuracy. Academia mainly studies problems of this kind from the point of view of being with or without detectors [24–27]. Although sliding mode control can be passively fault-tolerant without a detector [28], its fault-tolerant response is not sensitive enough, and its reconstruction accuracy is not high. Therefore, it is often combined with adaptivity, an observer, and other technologies. There is also a zero-order hold mechanism for event-triggering to compensate for faults while reducing communication resources [29,30]. In addition, the remote control signal of the robot is calculated and sent by the wireless upper control terminal, but the host may have some problems, such as Trojan horse implantation, redundant data interference, illegal attack, transmission noise, and so on. This will lead to matching interference in the transmission information [31]. However, for when these effects lead to abnormal control, how to better carry out the soft compensation of the control side is the focus of this paper.

To sum up, it can be known that reducing the parameter-tuning complexity of sliding mode reaching-law control to reduce chattering and ensure reaching efficiency is a direction that needs to be improved in the research of super-twisted reaching laws. The physical limitations of the robot lead to the integration-performance redundancy of unutilized residual errors on the traditional sliding mode surface, which is also challenging. In addition, how to better compensate for the abnormal control problems caused by signal interference, network attacks, and faults is also very important. Therefore, a new type of sliding surface was designed. The dynamic acceleration characteristics of the sliding mode are considered to improve the super-distortion reaching law, the nonlinear saturated filtering fault-tolerant mechanism is used to fit the abnormal information, and the adaptive neural network technology is used to fit and compensate. Finally, the stable control of artificial intelligence transportation robots was realized and event-triggered. The main innovations of this paper are as follows:

1. The integral processing easily produces the problem of stable error, but the error state is bounded; that is, the redundant information can be said to be bounded by the state, the reference trajectory, and information beyond the limit. The elimination of redundant information is limited to the bounded range, which effectively reduces the problem of stable error. Using the function of eliminating redundant information of residual error by integral term, the integral saturation mechanism is designed. While avoiding integral saturation, the redundant information of residual error, which can be offset by the maximum control performance, is removed. Compared with the ordinary integral sliding mode control, it will improve the controllable stability of sliding mode control in the case of fault tolerance and saturation.

2. The speed of traditional super-twist near the sliding mode surface is larger than that of the sliding mode dynamic method in this paper. From the angle of approaching dynamics, a better approaching state can be obtained from a lower speed, so there is a better buffeting suppression effect than traditional super-twist, and the parameter adjustment

is also simpler. The control method overcomes the shortcomings of the traditional super-twist reaching law, such as the complex adjustment of the parameters and the tendency to increase the buffeting when approaching the instantaneous mutation. Considering the acceleration dynamic characteristics of the reaching law, the nonlinear, variable damping reaching law is designed to reduce the change rate of the switching interval between the reaching stage and the sliding stage of the sliding mode, thereby improving the buffeting weakening ability and the reaching efficiency.

3. When solving the problem of the control signal being attacked by data interference and the partial failure of the actuator, the low fitting accuracy of the control abnormal information and the difficulty of signal-data interference-attack peeling in the literature [30–32] are overcome. The nonlinear fitting factor was designed based on the virtual hypothesis of abnormal information, and the nonlinear saturation-fault-tolerant filtering mechanism was designed for the dynamic information of system state.

The rest of this paper is divided into four sections: Section 2 establishes the kinematic and dynamic motion models of the tracked underwater vehicle and sets out the preliminary knowledge; Section 3 contains four parts: The first part proposes a new, nonlinear, projection redundant, feedforward sliding mode surface and a new sliding mode reaching law. It also provides the theoretical comparison and proof of the advantages of the method. The second part describes the event-triggered mechanism. Then, in the third part, the new nonlinear fault-tolerant subsystem is proposed, and its effectiveness is demonstrated. In the fourth part, the kinematics and dynamics controller are designed. In the Section 4, the Simulink simulation is compared with the control system using a traditional sliding mode approach law to verify the effectiveness of the control scheme proposed in this paper. Section 5 gives the conclusion of this paper.

## 2. Model and Preliminaries

### 2.1. Artificial Intelligence Transportation Robot Model

The stability and safety of artificial intelligence transportation robots in container cargo transportation are important, so differential mobile robots are often used. On the other hand, the differential mobile robots have the characteristic of a nonholonomic constraint. According to reference [33], the kinematic and dynamic models of robot motion plane are as follows:

$$\dot{q}(k) = S(q(k))u(k) \tag{1}$$

$$\dot{u}(k) = \bar{M}^{-1}(q)[\bar{B}(q)\tau_l - F_m(\dot{q}) - \bar{\tau}_d] \tag{2}$$

where $\bar{M}(q) = S^T(q)M(q)S(q)$, $\bar{\tau}_d = S^T(q)\tau_d$, $\bar{B}(q) = S^T(q)B(q)$, and $\tau_{\max} \geq \|\tau\|$. $F_m(\dot{q})$ is dynamic uncertainty of robot model. $M(q) = \begin{bmatrix} m & 0 & -md\sin\phi \\ 0 & m & mdc\cos\phi \\ -md\sin\phi & m\cos\phi & md^2 + J \end{bmatrix}$;

$C(q,\dot{q}) = \begin{bmatrix} 0 & 0 & -\dot{m}md\cos\phi \\ 0 & 0 & -\dot{m}md\sin\phi \\ 0 & 0 & 0 \end{bmatrix}$; $M(q) = \begin{bmatrix} m & 0 & -md\sin\phi \\ 0 & m & mdc\cos\phi \\ -md\sin\phi & m\cos\phi & md^2 + J \end{bmatrix}$;

$C(q,\dot{q}) = \begin{bmatrix} 0 & 0 & -\dot{m}md\cos\phi \\ 0 & 0 & -\dot{m}md\sin\phi \\ 0 & 0 & 0 \end{bmatrix}$; $T = \begin{bmatrix} T_L \\ T_R \end{bmatrix}$; $B(q) = \begin{bmatrix} \frac{\cos\phi}{r} & \frac{\sin\phi}{r} & -\frac{b}{r} \\ \frac{\cos\phi}{r} & \frac{\sin\phi}{r} & \frac{b}{r} \end{bmatrix}$; $A^T(q) =$

$\begin{bmatrix} \sin\phi \\ -\cos\phi \\ -d \end{bmatrix}$; $\lambda = -m(\dot{x}\cos\phi + \dot{y}\sin\phi)\dot{\phi}$; $\tau_d = \begin{bmatrix} T_{d1} \\ T_{d2} \\ T_{d3} \end{bmatrix}$. $\lambda$ is the Lagrange dynamics' dykoll coordinates to kinetic multipliers of generalized coordinates. The linear velocity

and angular velocity matrix is $u = \begin{bmatrix} v \\ w \end{bmatrix}$. The kinematic model's coefficient matrix

$$s(q) = \begin{bmatrix} \cos\theta & d\sin\theta \\ \sin\theta & -d\sin\theta \\ 0 & 1 \end{bmatrix}.$$

### 2.2. Mathematical Model of Abnormal Control

The data transmission network from controller to actuator is vulnerable to random noise interference and data-injection interference. In addition, the actuator of the artificial intelligence transportation robot will have a physical saturation limitation, so it is necessary to carry out saturation fitting in advance. Here, the input nonlinear fitting model is introduced. From Equation (2), the true actuator input of using hyperbolic tangent function to fit the saturation characteristics of robot actuators has the following form:

$$\tau_l(k) = A_\tau \tau_{\max} \tanh(\tau(k) + A_D) \tag{3}$$

where the $\tau(k)$ is the controller-calculated online signal. $A_r$ is an input partial fault with coupling characteristics. $A_D$ is a kind of data interference network attack with concealment characteristics. $A_D = \Im A_\Im$, $\Im$ has the characteristics of an independently distributed Bernoulli sequence with a value of $\{0\}$ or $\{1\}$. The $A_\Im$ is a virtual interference data value of the control signal caused by the attack.

### 2.3. RBF Neural Network Fitter

This is inspired by the paper [34]. The radial basis function neural network (RBFNN) approximation [35] is cited. As shown in Figure 2, a RBF neural network is a three-layer neural network because of the nonlinear characteristics of system uncertainty. If there exists an $m$-dimensional compact set $\Xi^m \subseteq R^m \to R$ and there is an unknown nonlinear function $f(Q)$ with initial value 0 defined on $\Xi^m$, the RBF approximator (4) is used to fit the dynamic values of $f(Q)$.

$$f(Q) = W^{*\,\mathrm{T}}Z(Q) + e_Z(Q), \quad \forall Q \in \Xi^m \tag{4}$$

where $e_z(Q)$ is the bounded RBF fitting error, which is defined on the compact set $\Xi^m$. $|e_z(Q)| \leq \bar{e}_z$, and $\bar{e}_z$ is the maximum nuclear distance. To improve the nonlinear local approximation ability, the Gaussian function $Z(Q) = \exp\left((Q-\kappa)^{\mathrm{T}}(Q-\kappa)/-l^2\right)$ is selected as the smooth kernel function. The $\kappa$ is approaching the center column distance vector. The $l$ is a varying constant value. $*$ is the order-$m$ dimensional weight row vector, which is optimally fitted: as

$$W^* = \arg\left(\min_{\tilde{W}} \left\{ \sup_{Q \in E^m} \left|\hat{W}^- Z(Q) - f(z)\right| \right\}\right) \tag{5}$$

where $\hat{W}$ is the minimum estimate of $W^*$ that optimizes $E_f(Q)$.

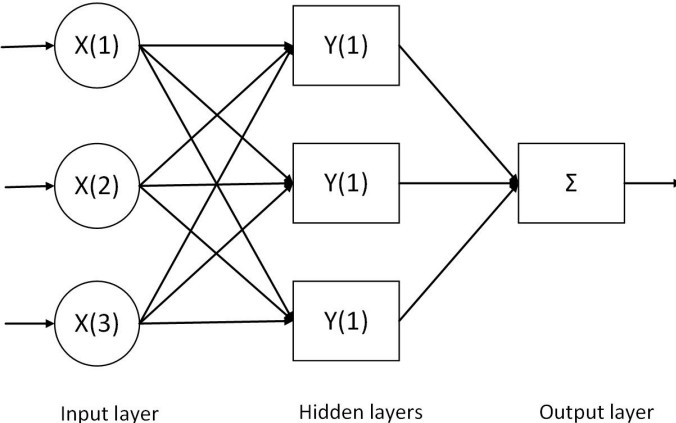

**Figure 2.** RBF neural network diagram.

The weight parameters in the online approximation process of the neural network approximator have been identified using parameter-adaptive technology, which is different from other neural networks [36] that generally need offline training. This method is based on the adaptive RBF neural network method in the paper [37], combined with the minimum parameter-learning method to perturbate the model's parameters caused by the disturbance and the neural network's weight parameters for online adaptation, so the weight matrix adjustment is automatically adjusted by the adaptive law. There are differences in the use of control processes in other areas.

The RBF neural network used in this paper is based on the ideas in the paper [35], and it is used as an online universal approximator, as a regression fitter, and it is fitted online with the data, so it is different from the general neural network, and this article adds the minimum parameter-learning method, as described in the paper [37]. The complexity and calculation time are adjusted with adaptive changes in the control process, and there is no need to train in advance when the uncertainty approximator of the control system, the activation function, can be set and the weight matrix can be determined to achieve universal approximation [35]. Therefore, taking advantage of the universal approximation property of RBF and the absence of a need for training in advance, the uncertainty caused by model dynamics and network attacks can be approximated nonlinearly.

*2.4. Preliminaries*

**Lemma 1.** *For the system (1), when $x^* \in \mathbb{R}$ and $x^* \neq 0$, if Lyapunov function $V(x^*) > 0$ exists, Lyapunov condition of the finite-time stability can be given as [30]*

$$\dot{V}(x^*) + \beta_1 V(x^*) + \beta_2 V^*(x^*) \leq 0 \tag{6}$$

*where $\beta_1 > 0, \beta_2 > 0$, and $0 < \kappa < 1$, so the system is globally finite-time stable, and the stable time depending on the initial state $e_0$ is given as*

$$T_V = \ln\left(\left(\beta_2 \cdot V^{1-\kappa}(e_0) + \beta\right)/\beta_2\right)/(\beta_1 - \beta_1\kappa) \tag{7}$$

*The relevant proof is shown in reference [30].*

**Lemma 2.** *According to the Cauchy–Schwarz inequality, for any number $a_i$ and $b_i (i = 1, 2, \ldots, n)$, we can know*

$$\left(\sum_{i=1}^{n} a_i b_i\right)^2 \leq \left(\sum_{i=1}^{n} a_i^2\right)\left(\sum_{i=1}^{n} b_i^2\right) \tag{8}$$

*and any $0 < l < 1$, there exists*

$$\left(\sum_{i=1}^{n}|a_i|\right)^t \leq \sum_{i=1}^{n}|a_i|^t \tag{9}$$

**Lemma 3.** *For any $b_c > 0$ and $z_t \in \mathbb{R}$, $\tanh(\cdot)$ has the following property:*

$$0 \leq |z_t| - z_t \tanh(z_t/b_c) \leq 0.2785b_c \tag{10}$$

**Assumption 1.** *The unknown bounded low frequency time-varying disturbance $\bar{\tau}_d$ is $\|\bar{\tau}_d\| \leq \hat{\tau}_d$. The initial system state errors, $u_e(0)$ and $q_e(0)$, are defined on a compact set, and it is assumed as $\|u_e(t)\| \leq \bar{u}_e$ and $\|q_e(t)\| \leq \hat{q}_e$.*

**Assumption 2.** *To the limited range of the freight yard, the range of motion and the desirably reference trajectory of the robot are bounded . The desirably reference trajectory $q_r = \begin{bmatrix} x_r & y_r & \theta_r \end{bmatrix}^T$, and reference positive scalar speed $u_r = \begin{bmatrix} v_r & w_r \end{bmatrix}^T$. Their derivatives are smooth and bounded.*

**Assumption 3.** *The system (6) is a controllable system that satisfies Lemma 2. For facilitate matrix operation, all constant terms are in the form of a diagonal matrix.*

### 3. Controller Design

The virtual kinematic subsystem is designed to analyze velocity state in this section. It can obtain tracking position of artificial intelligence transportation robots. Based on characteristics of contaminated velocity state signals, the fault-tolerant filtering subsystem is designed. Then, the nonlinear sliding mode surface is designed to improve the robustness of the controller by establishing a nonlinear bounded state space and combined with the reaching law to reduce the vulnerability of faults and contaminated communications. Figure 3 is the schematic diagram of artificial intelligence transportation robot trajectory-tracking control flow.

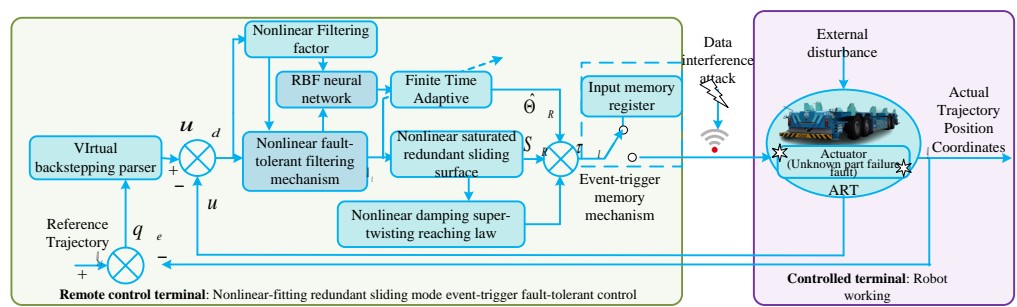

**Figure 3.** Schematic diagram of artificial intelligence transportation robots with a trajectory-tracking control flow.

*3.1. Nonlinear Saturation Fault-Tolerant Filtering Mechanism*

According to the system (1), the kinematic position error is defined as

$$q_e = \begin{bmatrix} x_e \\ y_e \\ \theta_e \end{bmatrix} = \begin{bmatrix} \cos\theta & \sin\theta & 0 \\ -\sin\theta & \cos\theta & 0 \\ 0 & 0 & 1 \end{bmatrix} \cdot \begin{bmatrix} x_r - x \\ y_r - y \\ \theta_r - \theta \end{bmatrix} \tag{11}$$

According to the characteristics of virtual backstepping and kinematic spatial, the method of velocity motion control at low latitudes of dynamics is used to design a virtual-kinematics control law as the dynamic desire value (12).

$$u_d = \begin{bmatrix} v_r \cos \theta_e - w\theta_e + \lambda_2(x_e - d + d\cos\theta_e) \\ w_r + \lambda_1^{-1} v_r(\vartheta_1(y_e + \theta_e) + (d\vartheta_1 + \lambda_1\vartheta_2)\sin\theta_e) \end{bmatrix} \tag{12}$$

where $u_r = [v_r, w_r]^T$ is the velocity of the reference trajectory.

Combined with the system state information, the error is designed as

$$U_e = \begin{bmatrix} v_e \\ \theta_e \end{bmatrix} = u_d - u \tag{13}$$

**Remark 1.** *The system input is contaminated by GPS sensor failure, unreliable signal, and actuator failure in the system, which will result in an unknown input. After this kind of input is executed, it will not be able to achieve stability control. Therefore, the nonlinear factor $\eta_\tau = \frac{A_\tau}{1+\tanh(\tau)\tanh(A_d)}$ is involved in constructing the saturation-fault-tolerant mechanism. The nonlinear saturation-fault-tolerant filtering mechanism $\eta_f(k) = 1 - \eta_\tau + \eta_\tau \tanh(A_d)$ denotes that the decoupling form about the unknown influence of the signal is dealt with nonlinearly. According to error Equation (2), the fault-tolerant saturation filtering dynamic subsystem is designed as $\dot{u} = \bar{M}^{-1}(q(k))[\bar{B}(q(k))\tau_l(k) - \bar{F}_m(\dot{q}(k)) - \bar{\tau}_d(k)] + \eta_f(k)$. The $\bar{F}_m(\dot{q}(k)) = F_m(\dot{q}(k)) + \tau_{\max}\bar{M}(q(k))\eta_f(k)$.*

*This mechanism makes use of unknown information of attack and fault loss to form a nonlinear virtual hypothesis. According to this hypothesis, signal attack and fault features can be extracted better, and the fault-tolerant and adaptive ability can be improved further according to saturation analysis method in the literature [32]. The filtering dynamic error is*

$$\dot{E}_R(k) = \dot{u}_d(k) - \left( \bar{M}^{-1}(q(k))[\bar{B}(q(k))\tau_l - \bar{F}_m(\dot{q}(k)) - \bar{\tau}_d] \right) - \eta_f(k) \tag{14}$$

*3.2. Design of Nonlinear-Fitting, Redundant, Sliding Mode, Event-Trigger Fault-Tolerant Control*

**Step 1.** A new type of a nonlinear, saturated, redundant sliding surface (NSRSMS). The NSRSMS is denoted as:

$$S_R(k) = \dot{E}_R(k) + \gamma_a E_R(k) + B_0(E_R(k)) - B_l(E_R(k)) \tag{15}$$

where $B_l(E_R(k)) = \int \tanh(E_R(t)) \ln\left[ \frac{B_\psi(E_R(l))\exp(1)+1}{1+\exp(1)} \right] dl$, and $B_\psi(E_R(k)) = \beta_a E_R(k) \tanh (E_R(k))$.

The $\beta_a = \mathrm{diag}\{\beta_{a1}, \beta_{a2}\}$ is positive permanent diagonal function. The $B_0(E_R(k)) = -[\gamma_a E_R(0) - l(E_R(k))]$ is initial global approach term to ensure the global mode of SMC.

**Remark 2.** *When the error is designed on a SMS, the maximum error that can be eliminated at a single time has the characteristic of saturation because of the physical limitation of the control ability of the controlled robot. Therefore, the redundant information of saturation is used to give full play to the control performance, prevent the control performance from overshoot, and design a nonlinear mechanism $B_l(E_R(k))$ to improve the integral saturation. The characteristics of NSRSMS and linear SMS $S_I(k) = \dot{E}_R(k) + \int E_R(l)dl + E_R(k) + B_0(E_R(k))$ are shown in Figure 4, where $\int E_R(l)dl$ is y-axis and $E_R(k)$ is the x-axis.*

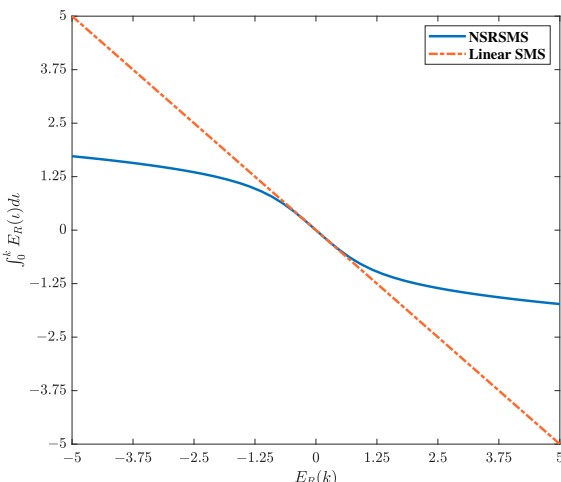

**Figure 4.** Comparison of NSRSMS and linear sliding mode surface.

**Remark 3.** *The control performance error struggles to meet the error overshoot when the error is large; that is to say, the error of the expected performance is relatively easy to control, and if the error exceeds a certain expected region, it will need to be compensated step by step. It can be seen in Figure 4 that the saturation term is designed and applied to the sliding surface, the saturated information is regarded as redundant information, the filtered saturated information is filtered directly, and the expectation of a certain error is used as the fault-tolerant information control point. It will be more conducive to the realization of fault tolerant control and reduce the waste of control performance.*

*In addition, to avoid control instability caused by singularity in practical engineering, the singularity of the sliding surface is verified through Equation (16).*

$$\dot{S}_R = \ddot{E}_R(k) + \gamma_a \dot{E}_R(k) + \dot{B}_0(E_R(k)) - B_\gamma \tag{16}$$

*where* $B_\gamma = \tanh(E_R(t)) \left[ \ln \left( B_\psi(E_R(t)) \exp(1) + 1 \right) + \ln(1 + \exp(1)) \right]$, *and we can know that NSRSMS does not contain singularities in the control process.*

**Step 2.** Nonlinear-damping, super-twisting reaching law (NDSTRL) considering acceleration.

Although the redundant information of the integral saturated sliding mode surface can be used to improve the control accuracy and reduce the residual error, if there is a sudden instantaneous error, the buffeting problem can not be ignored, so the NDSTRL is designed according to the traditional STRL. The NSTRL is denoted as

$$\dot{S}_r = -\gamma_r \chi(k) \operatorname{sign}\left(S_R^T\right) - \gamma_k S_R \tag{17}$$

where $\chi(k) = \tanh\left( \|S_R(k)\|^{\frac{3}{2}} \right) \|S_R(k)\|^{\frac{1}{2}}$ shows nonlinear time-varying gain. The STRL denotes $\dot{S}_s = -\gamma_{rs} \|S_R(k)\|^{\frac{1}{2}} \operatorname{sign}\left(S_R^T\right) - \gamma_{ks} S_R$, and ERL is defined as $\dot{S}_E = -\gamma_{rE} \operatorname{sign}\left(S_R^T\right) - \gamma_{kE} S_R$. Figure 5 shows NDSTRL, STRL, and ERL with coefficients equal to one.

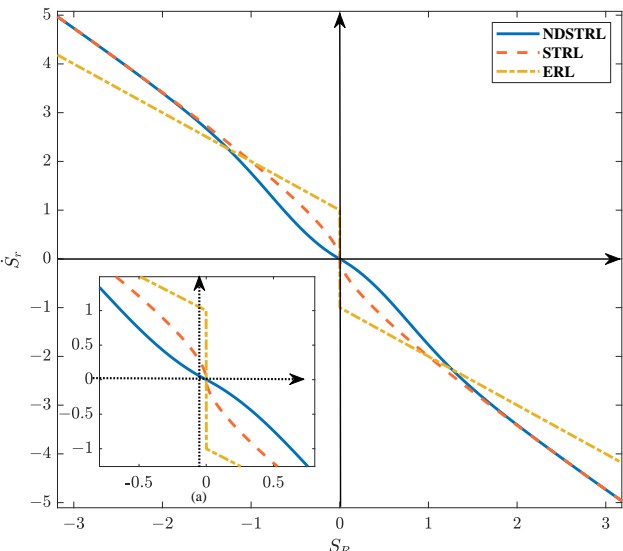

**Figure 5.** Comparison of NDSTRL and STRL.

**Remark 4.** *From Figure 5, we can see that the STRL can effectively regulate the reaching speed according to the system state value. When the reaching speed reaches the SMS, the reaching speed will be decelerated smoothly. The sudden deceleration at the break point of ERL (see the ERL in Figure 5a), which can not avoid chattering, will increase the chattering amplitude. Figure 5a shows that the acceleration of the NDSTRL decreases gradually, and the speed is adjusted gradually. The acceleration of SMC reaching dynamics are better controlled, which is more conducive to reducing buffeting. Additionally, even better, in the case of a large error, the speed of sliding mode will not be reduced.*

**Remark 5.** *In addition, we can see in Figure 6 that the NDSTRL acceleration gain has the effect of adjusting the acceleration trend. The status is closer to that of the SMS; the gain is more, which will slow down the velocity trend (see Figure 6a). Moreover, the reaching speed can be dynamically accelerated when the error is large, and speed gain does not need to be adjusted (see Figure 6b).*

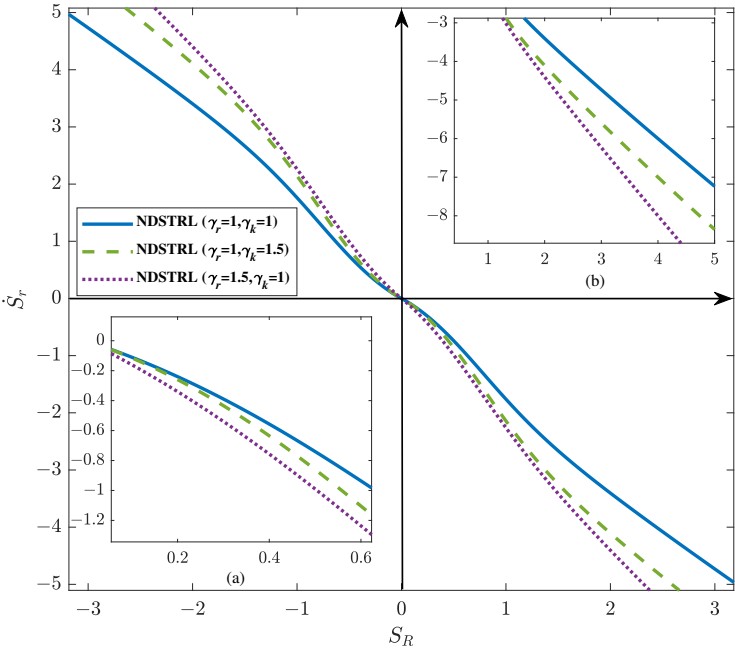

**Figure 6.** Comparison of NSTRL and STRL.

**Step 3.** Controller based on event-triggered memory input mechanism.

Under the influence of unexpected situations such as network attacks and faults, saving communication resources can reduce the instability of network control to a certain extent, so an online memory input event-triggered mechanism was designed (shown in Equation (18)). This mechanism allows the controlled robot to complete stable control without achieving the ideal control precision, which can also help to achieve stable control with a certain level of precision under network attacks.

$$\|T_l(k)\| \geqslant \lambda\|\tau_l(k_l)\| \tag{18}$$

where $\vartheta|\tau_l(t_k)|$ is the memory input with trigger weight gain, and $\tau_l(t_k)$ is the input that meets the trigger condition. $T_l(k) = \tau_l(k) - \tau_l(k_l)$ shows the dynamic characteristics and the input, and contains the state information of the robot affected by faults and attacks. When the input value of online calculation satisfies Equation (18), $\tau_l(k) = \tau_l(k_l), \forall k \in [k_l, k_{l+1})$, it denotes the input affected by the zero-order retention effect of the robot installed in advance.

Using Equation (17) and Equations (21)–(39), one can obtain the control law (Equation (19)) and adaptive law (Equation (20)).

$$\tau_l(k) = \tau_c(k) + \tau_\varsigma(k) \tag{19}$$

$$\dot{\hat{\Theta}}_R = \frac{1}{4r_s} \tanh\left(\frac{S_R}{D}\right) S_R - b\hat{\Theta}_R + r_s r_\Theta \|S_R\|^2 \varphi^4(Z_R) \tag{20}$$

where $\tau_\varsigma(k)$ denotes an adaptive nonlinear fault-tolerant filter control mechanism, and $\tau_c(k)$ is the NDSTRL distance. $\tau_c(k) = -(\gamma_a\tau_{\max}\tilde{B})^{-1}G_c\dot{S}_r + \gamma_a\dot{u}_d$ and $\tau_\varsigma(k) = (\gamma_a\tau_{\max}\tilde{B})^{-1}\left(\frac{1}{4r_s}\tanh\left(\frac{S_R}{D}\right) + r_s r_\Theta S_R \varphi^4(Z_R)\right)\hat{\Theta}_R.$ $r_\Theta > 0, r_s > 0, Z_1 = \left[H_1^{\mathrm{T}}, \dot{u}_d^{\mathrm{T}}, \varsigma^{\mathrm{T}}\right]^{\mathrm{T}}, \Theta_R = \max\{\|\tau_{\max}E_r\|, \|\psi^T\|, \|\tau_{\max}E_f\|, \|\bar{\tau}_D\|\}, H_1 = \ddot{E}_R(k) + \dot{B}_0(E_R(k)),$ and $\varphi(Z_R) = \|D_r\|(2\|\tilde{B}^T(q)\| + \|\tilde{B}^T(q)\tau_{\max}\phi(u)\| + \|D_r\|^{-1}\|\ddot{E}_R(k) + \dot{B}_0(E_R(k))\|).$

*3.3. Theoretical Proof*

To verify the stability of the controller, the effectiveness of the kinematic virtual controller, dynamic controller and event-triggered controller will be verified theoretically.

**Proof.** Theoretical proof of the virtual-kinematics control law.

Next, take the Lyapunov function to prove its stability:

$$V_q = \frac{1}{2}S_x^2 + \frac{1}{2}S_y^2 + S_\theta \tag{21}$$

where $S_x = x_e + d\cos\theta_e + d, S_y = y_e + d\sin\theta_e + \theta_e, S_\theta = k_1(1 - \cos\theta_e), k_1$ and $k_2$ are positive parameters. After the first order guidance, one can obtain

$$\dot{V}_q = S_x\dot{S}_x + S_y\dot{S}_y + \dot{S}_\theta \tag{22}$$

so we obtain

$$\dot{V}_q = -\lambda_2(x_e - d + d\cos\theta_e)^2 - \alpha\frac{v_r}{\lambda_1}(y_e + d\sin\theta_e + \theta_e)^2 \leq 0 \tag{23}$$

As (23) is negative, the system tends to be stable, and it is also proved in [26].

$$u_d = \begin{bmatrix} v_r\cos\theta_e - w\theta_e + \lambda_2(x_e - d + d\cos\theta_e) \\ w_r + \lambda_1^{-1}v_r(\vartheta_1(y_e + \theta_e) + (d\vartheta_1 + \lambda_1\vartheta_2)\sin\theta_e) \end{bmatrix} \tag{24}$$

☐

**Proof.** Theoretical proof of the dynamic controller.

The Lyapunov function $V_{R1} = \frac{1}{2}S_R^T S_R$ is designed, and the first derivative of $V_{R1}$ with sampling time can be obtained:

$$
\begin{aligned}
\dot{V}_{R1} &= S_R^T \dot{S}_R \\
&= S_R^T \big[ D_r \big( \dot{u}_d - \tilde{B}(q)\eta_\tau \tau_{\max}(\tanh(\tau) + \tanh(A_D)) \\
&\quad + \bar{M}^{-1}(q)\bar{F}_m(\dot{q}) - \eta_f + \bar{\tau}_D \big) + \ddot{E}_R(k) + \dot{B}_0(E_R(k)) - B_\gamma \big]
\end{aligned}
\tag{25}
$$

where $\gamma_a = D_r, \tilde{B}(q) = \bar{M}^{-1}(q)\bar{B}(q), \bar{\tau}_D = \bar{M}^1(q)\bar{\tau}_d$.

By using the saturation analysis mechanism (see Remark 1), the multiplicative fault is nonlinearized and projected, and one can obtain:

$$
\begin{aligned}
\dot{V}_{R1} &\le S_R^T \big[ D_r \big( \dot{u}_d - \tilde{B}(q)\tau_{\max}(\tanh(\tau) + (1 - \eta_\tau + \eta_\tau \tanh(A_D))) \\
&\quad + \bar{M}^{-1}(q)\bar{F}_m(\dot{q}) - \eta_f + \bar{\tau}_D \big) + \ddot{E}_R(k) + \dot{B}_0(E_R(k)) - B_\gamma \big]
\end{aligned}
\tag{26}
$$

According to $\eta_f = 1 - \eta_\tau + \eta_\tau \tanh(A_d)$, the nonlinear fitting characteristics of neural network are used to fit the fault, attack, and dynamic nonlinearity. One can obtain

$$
\begin{aligned}
\dot{V}_{R1} &\le S_R^T \big[ D_r \big( \dot{u}_d - \tilde{B}(q)\tau_{\max}(\tau - E_\tau) + \dot{B}_0(E_R(k)) - B_\gamma \\
&\quad + \tau_{\max}\tilde{B}^T(q)\big( \psi^T \phi(u) + E_f \big) + \bar{\tau}_D \big) + \ddot{E}_R(k) \big]
\end{aligned}
\tag{27}
$$

where $E_\tau = \tau - \tanh(\tau), F_D(\dot{q}) = \tau_{\max}^{-1}\bar{B}^{-1}(q)F_m(\dot{q}) + \eta_f$. According to the above formula, we can obtain:

$$
\begin{aligned}
&\big\| S_R^T \big\| \big( \| \tau_{\max}E_\tau \| \big\| D_r\tilde{B}^T(q) \big\| + \big\| \psi^T \big\| \big\| D_r\tilde{B}^T(q)\tau_{\max}\phi(u) \big\| + \big\| \tau_{\max}E_f \big\| \big\| D_r\tilde{B}^T(q) \big\| \\
&+ \| \bar{\tau}_D \| \| D_r \| + \big\| \ddot{E}_R(k) + \dot{B}_0(E_R(k)) \| \big) \le \big\| S_R^T \big\| \Theta_R \varphi(Z_R)
\end{aligned}
\tag{28}
$$

From this, we can obtain

$$
\dot{V}_{R1} \le \big\| S_R^T \big\| \Theta_R \varphi(Z_R) - S_R^T D_r \big[ \dot{u}_d - \tilde{B}(q)\tau_{\max}\tau \big]
\tag{29}
$$

The global Lyapunov function is denoted as

$$
V_R = V_{R1} + \frac{1}{2}\tilde{\Theta}_R^2
\tag{30}
$$

where $\tilde{\Theta}_R = \Theta_R - \hat{\Theta}_R$, and $\dot{\tilde{\Theta}}_R = -\dot{\hat{\Theta}}_R$. The time derivative of $V_u$ is

$$
\dot{V}_u = \dot{V}_{R1} - \tilde{\Theta}_R^T \dot{\hat{\Theta}}_R \le \big\| S_R^T \big\| \Theta_R \varphi(Z_R) - S_R^T D_r \tau_{\mathrm{ma}} \tilde{B}^T(q)\big( \tau_c(k) + \tau_\xi(k) \big) - \tilde{\Theta}_R^T \dot{\hat{\Theta}}_R
\tag{31}
$$

By substituting Equations (19)–(20) into Equation (31), one can obtain

$$
\begin{aligned}
\dot{V}_R &\le \big\| S_R^T \big\| \Theta_1 \varphi(Z_R) + S_R^T \dot{S}_r + S_R^T \bigg[ -\frac{1}{4r_s} \tanh\bigg( \frac{S_R}{D} \bigg) \hat{\Theta}_R \\
&\quad - r_s r_\Theta S_R \hat{\Theta}_1 \varphi^4(Z_R) \bigg] - \tilde{\Theta}_1^T \bigg[ \frac{1}{4r_s} \tanh\bigg( \frac{S_u}{D} \bigg) S_u - b\hat{\Theta}_R \\
&\quad + r_s r_\Theta \| S_R \|^2 \varphi^4(Z_R) \bigg]
\end{aligned}
\tag{32}
$$

According to Young's inequality, we can obtain the following:

$$\left\|S_R^{\mathrm{T}}\right\|\Theta_R\varphi(Z_R) \leq r_s\left\|S_R^{\mathrm{T}}\right\|\Theta_R\varphi^2(Z_R) + \frac{\left\|S_R^{\mathrm{T}}\right\|\Theta_R}{4r_s} \tag{33}$$

Substitute Equation (33) into Equation (32) as

$$\begin{aligned}
\dot{V}_R &\leq S_R^{\mathrm{T}}\dot{S}_r - \frac{1}{4r_s}S_R^{\mathrm{T}}\tanh\left(\frac{S_R}{D}\right)\Theta_R + r_s\left\|S_R^{\mathrm{T}}\right\|\Theta_R\varphi^2(Z_R) \\
&\quad - \tilde{\Theta}_R^{T}\left(-b\hat{\Theta}_R + r_s r_\Theta\|S_R\|^2\varphi^4(Z_R)\right) + \frac{\left\|S_R^{\mathrm{T}}\right\|\Theta_R}{4r_s} \\
&\quad - r_s r_\Theta\|S_R\|^2\hat{\Theta}_R\varphi^4(Z_R)
\end{aligned} \tag{34}$$

Using Young's inequality, we can obtain

$$r_s\Theta_R\left\|S_R^{\mathrm{T}}\right\|\varphi^2(Z_R) \leq r_s r_\Theta\Theta_R\left\|S_R^{\mathrm{T}}\right\|^2\varphi^4(Z_R) + \frac{r_s\Theta_R}{4r_\Theta} \tag{35}$$

Further, using Equation (35), $\tilde{\Theta}_1 = \Theta_1 - \hat{\Theta}_1$ can be rewritten as

$$\dot{V}_R \leq S_R^{\mathrm{T}}\dot{S}_r - \frac{1}{4r_s}S_R^{\mathrm{T}}\tanh\left(\frac{S_R}{D}\right)\Theta_R + \frac{\Theta_R}{4r_s}\left\|S_R^{\mathrm{T}}\right\| + b\tilde{\Theta}_R^{T}\hat{\Theta}_R + \frac{r_s\Theta_R}{4r_\Theta} \tag{36}$$

According to Lemma 3 and $\tilde{\Theta}_R^{T}\hat{\Theta}_R \leq \tilde{\Theta}_R^{T}(\Theta_R - \tilde{\Theta}_R) \leq \frac{1}{2}\Theta_R^2 - \frac{1}{2}\tilde{\Theta}_R^{T}\tilde{\Theta}_R$, we have the following inequality:

$$\begin{aligned}
\dot{V}_R &\leq S_R^{\mathrm{T}}\dot{S}_r - \frac{1}{4r_s}S_R^{\mathrm{T}}\tanh\left(\frac{S_R}{D}\right)\Theta_R + \frac{\Theta_R}{4r_s}\left\|S_R^{\mathrm{T}}\right\| + b\tilde{\Theta}_R^{T}\hat{\Theta}_R + \frac{r_s\Theta_R}{4r_\Theta} \\
&\leq S_R^{\mathrm{T}}\dot{S}_r - \frac{b}{2}\tilde{\Theta}_R^{T}\tilde{\Theta}_R + \frac{1}{4r_s}0.2785D\Theta_R + \frac{r_s\Theta_R}{4r_\Theta} + \frac{b}{2}\Theta_R^2
\end{aligned} \tag{37}$$

Using $\frac{1}{4}\left\|\tilde{\Theta}_R\right\| \leq \frac{b}{4}\left\|\tilde{\Theta}_R\right\|^2 + \frac{1}{16b}$, we can obtain

$$\begin{aligned}
\dot{V}_R &\leq S_R^{\mathrm{T}}\dot{S}_r - \frac{1}{4}\left(\tilde{\Theta}_R^{T}\tilde{\Theta}_R\right)^{\frac{1}{2}} - \frac{b}{4}\tilde{\Theta}_R^{T}\tilde{\Theta}_R + \frac{1}{4r_s}0.2785D\Theta_R \\
&\quad + \frac{r_s\Theta_R}{4r_\Theta} + \frac{b}{2}\Theta_R^2 + \frac{1}{16b}
\end{aligned} \tag{38}$$

According to Equation (38), we can obtain

$$\begin{aligned}
\dot{V}_R &\leq -\left(S_R^{\mathrm{T}}\left(\sqrt{2}\gamma_r(g_s - I_E)\right)^2 S_R\right)^{\frac{1}{2}} - S_R^{\mathrm{T}}\left(\gamma_r + \gamma_k - \frac{1}{2}I_E\right)S_R \\
&\quad - \frac{1}{4}\left(\tilde{\Theta}_R^{T}\tilde{\Theta}_R\right)^{\frac{1}{2}} - \frac{b}{4}\tilde{\Theta}_R^{T}\tilde{\Theta}_R + \circlearrowleft \\
&\leq -\eta_1 V_R - \eta_2 V_R^{\frac{1}{2}} + \circlearrowleft
\end{aligned} \tag{39}$$

where $\eta_1\left\{\lambda_{\min}\left(\gamma_r + \gamma_k - \frac{1}{2}I_E\right), \frac{\lambda_{\min}(b)}{4}\right\}$, $\eta_2 = \left\{\lambda_{\min}\left(\sqrt{2}\gamma_r(g_s - I_E)\right), \frac{1}{4}\right\}$, and $\circlearrowleft = \frac{1}{4r_s}0.2785D\Theta_R + \frac{r_s\Theta_R}{4r_\Theta} + \frac{b}{2}\Theta_R^2 + \frac{1}{16b}$. $\quad\square$

## 4. Simulation Results and Analysis

To verify the effectiveness and superiority of this control algorithm, a comparative simulation, SISMAEFC, was established by using a conventional integral SMC combined with the super-twisted sliding mode reaching law [38], and the algorithm was verified and analyzed by error-simulation results, sliding surface simulation results, and double-actuator-input simulation results (see Figures 7–10). In addition, because SISMAEFC does not use the improved nonlinear fault tolerance mechanism and does not reconstruct faults and attacks, it is compensated directly by adaptive algorithms, which is demonstrated by adaptive laws (Figure 11). All simulations in this section adopt control parameters as: $\gamma_a = \text{diag}\{45, 15.5\}$, $\beta_a = \text{diag}\{0.01, 0.01\}$, $G_c = \text{diag}\{1, 1\}$, $\tau_{max} = \text{diag}\{10, 10\}$, $\gamma_r = \text{diag}\{10, 40\}$, $\gamma_k = \text{diag}\{0.01, 0.1\}$, $r_s = \text{diag}\{0.02, 0.04\}$, $r_\Theta = \text{diag}\{0.01, 0.01\}$, $b = \text{diag}\{10.5, 10.5\}$, $D = \text{diag}\{1, 1\}$. According to the experimental model of the nonholonomic underactuated robot in the laboratory as the controlled object, the relevant parameters are: $m = 15\,\text{kg}$, $r = 0.05\,\text{m}$, $b = 0.5\,\text{m}$, $J = 5\,\text{kg} \cdot \text{m}^2$, $d = 0.05\,\text{m}$, $r = 0.05\,\text{m}$. The RBF$\cdot$NNs for H(Z) contain 15 nodes with centers evenly spaced in the range $[-3, 3] \times \ldots \times [-3, 3]$ and widths $\omega_l = 1.8(l = 1, \ldots, 15)$. The simulation interval is designed according to the sampling rate 100hz of the main control chip of the experimental robot. The total simulation time was 150 s. To verify the tracking control performances on different trajectories and the control stability under attack, the expected trajectory used a trapezoidal line with a combination of a straight line and a curve [39].

The position error and angular velocity error of the robot under the two algorithms change over time, and the position error under the algorithm designed in this paper can converge to zero quickly. From Figure 7, it can be observed that $x_e$ stabilized after about 4 s at the earliest stage, and $\phi_e$ stabilized at the latest at around 13 s. The whole system's position error could converge in about 15 s, and the stability of the curve was smooth relatively after the change in curvature after convergence. Although the error curve of SISMAEFC can converge, the curve shows obvious jitter, and SISMAEFC recovered to a stable state slowly when numerical fluctuation occurred, which was affected greatly by model uncertainty, and the convergence was not as good as that of the algorithm designed in this paper. Additionally, due to the network attack (the attack frequency is 30%), it can be seen from the local detail diagram that the SISMAEFC jitter is more obvious and the attack has a greater impact. It is worth noting that the error state under the NRSMEFC attack and affected by the fault is better, so we can see the attack and fault in this paper.

As shown in Table 1, the two parameters of the algorithm in this paper are smaller, and the MIAC is 18% lower than the minimum value and 45% lower than the maximum value of the comparison algorithm. The MISE is 22.6% lower than the minimum value and 11.4% lower than the maximum value of the comparison algorithm. Additionally, under the same attack, we can see from Figures 9 and 10 that the proposed algorithm is better.

**Table 1.** Quantitative analysis of the controller's control effect.

| Evaluation Criteria | MIAC | MISE |
|---|---|---|
| Algorithm in this paper | [0.90, 2.03] | [8.307, 8.844, 3.16] |
| Comparison algorithm | [1.107, 3.75] | [10.74, 10.13, 3.57] |

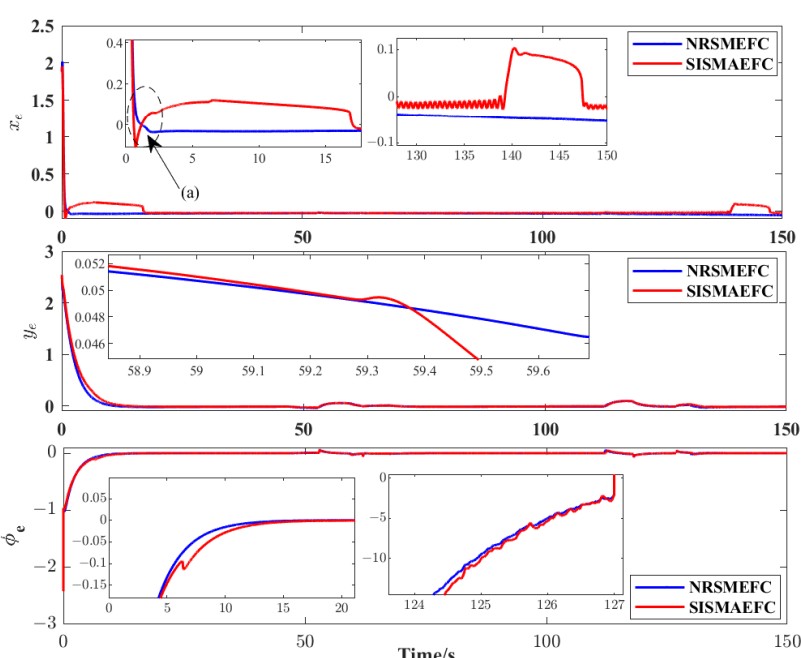

**Figure 7.** Comparison of system error.

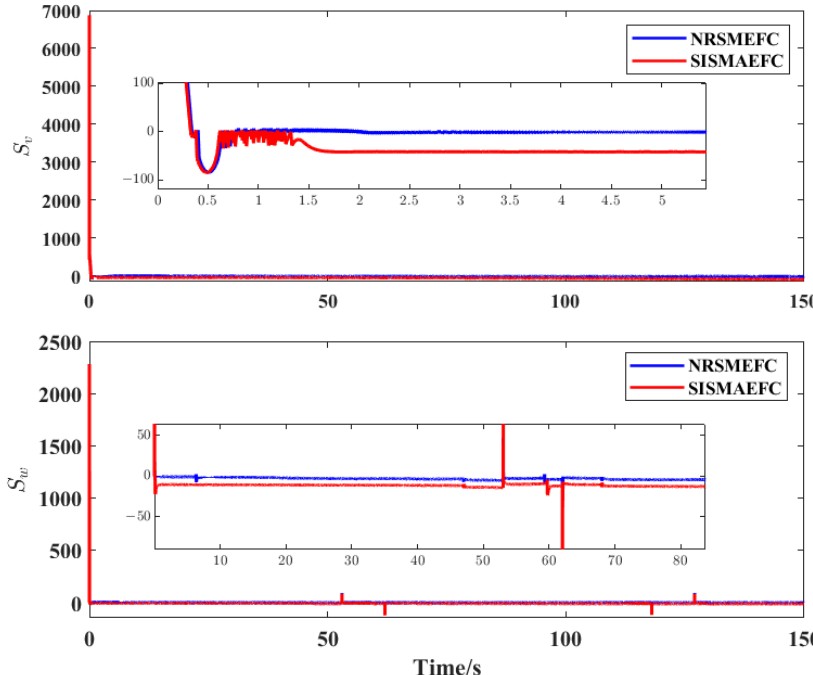

**Figure 8.** Sliding surface $S_R$ comparison.

As shown in Figure 8, when the surface of the traditional sliding surface changes abruptly, the influence of nonlinear terms will increase significantly near the sliding surface. The system's response changes, and the system's buffeting changes significantly. Due to the accelerated dynamic characteristics of the reaching law, the residual redundancy will appear when the system's error reaches the sliding mode surface. In this paper, the sliding surface is designed to make up for the deficiency of the traditional sliding surface under the integral saturation mechanism and nonlinear variable damping reaching law. It reduces the sharp change in the curvature of the sliding surface, thereby reducing the probability of

surface mutation. Thus, it eliminates the chattering phenomenon, makes up for the error redundancy, and improves the stability rate.

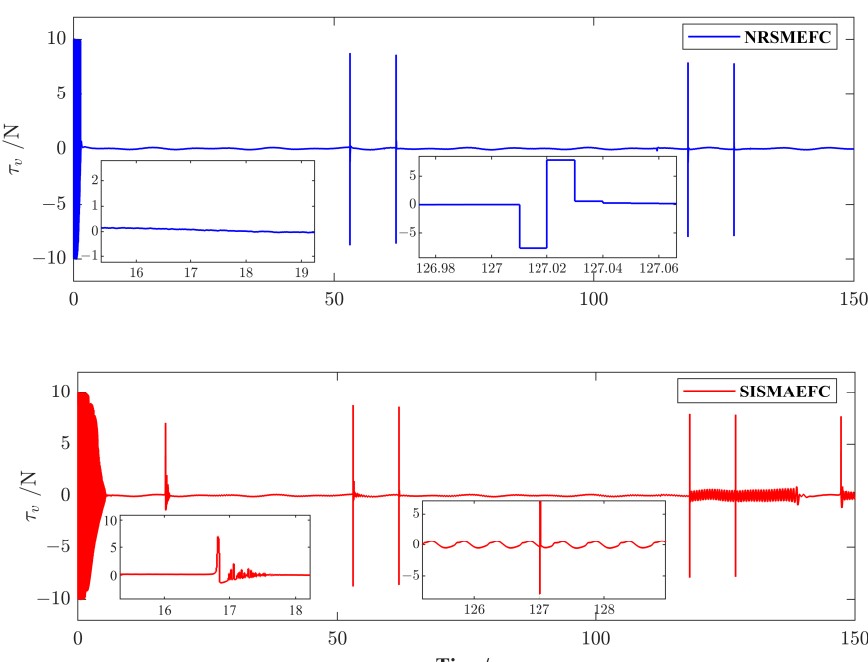

**Figure 9.** Differential coupling input $\tau_v$ comparison.

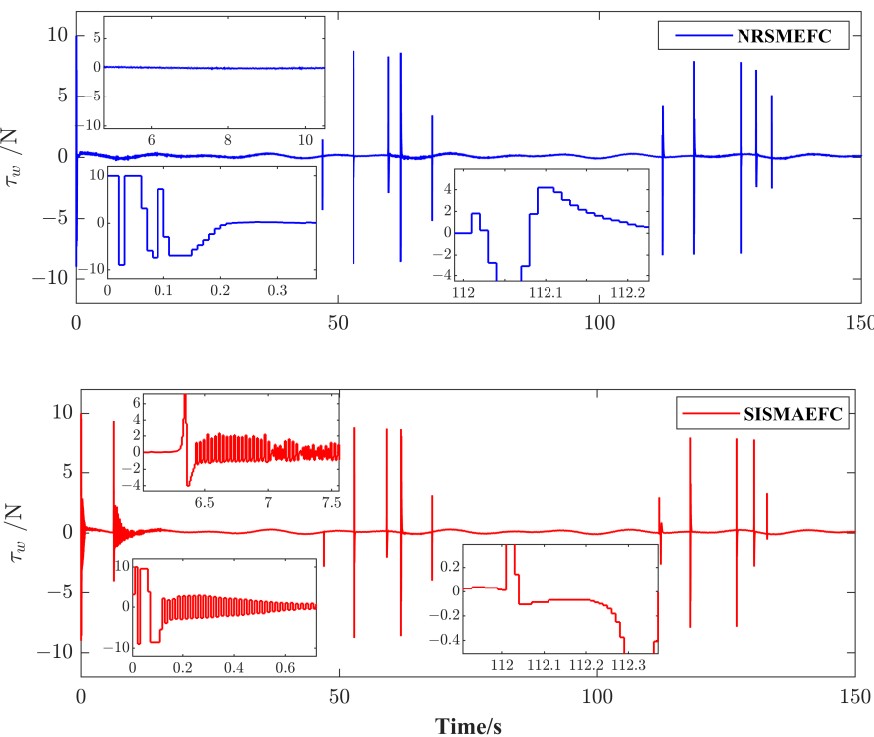

**Figure 10.** Differential coupling input $\tau_w$ comparison.

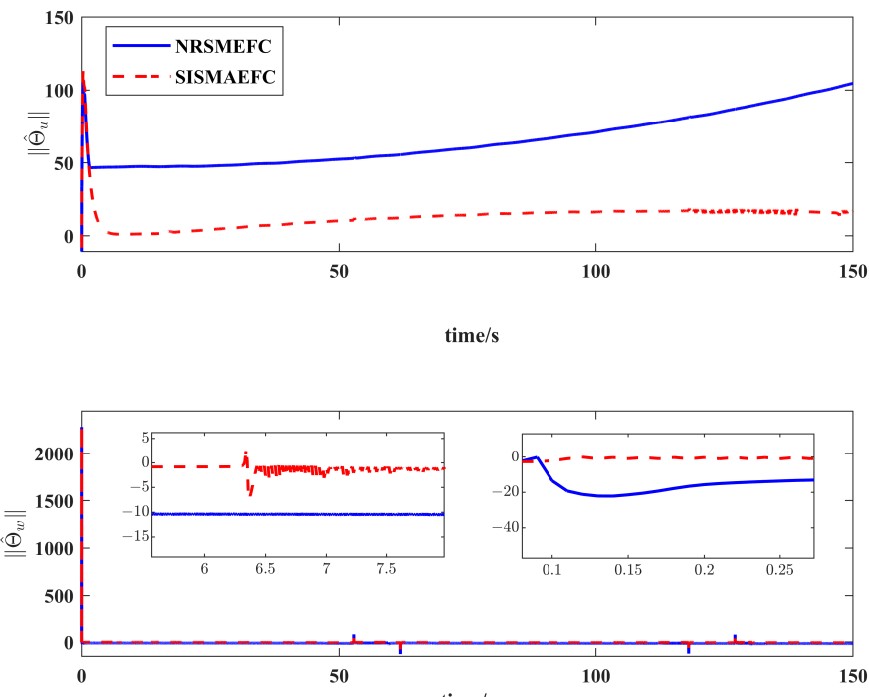

**Figure 11.** Comparison of compensation values of adaptive law.

From Figures 9 and 10, we can see that the control input of the algorithm designed in this paper does not fluctuate obviously under the attack and continues to be stable over time, but the control scheme shows severe chattering when the attack occurs. In Figure 11, we can see that the system cannot read the input signal correctly, which will affect the control effect. It can be seen in trigger times that the dynamic effect is better for the scheme of nonlinear fitting, followed by adaptive compensation by the controller in this paper. As the system's input may be subject to data errors, security attacks, system failures, and other problems, the system's input accuracy cannot be guaranteed. As can be seen in Figure 10, under the algorithm in this paper, the effect of event trigger is obviously better. The active transmission frequencies of the dual controller output channel of the proposed method and the comparison method are (8671, 6563) and (10,287, 11,733), respectively, which shows that the method in this paper has better control performance and is more stable state in cases of attacks and failures.

## 5. Conclusions

In this paper, a neural network, nonlinear-fitting, redundant, sliding mode event-trigger control system affected by abnormal input was designed. Firstly, according to the dynamic saturation input characteristics, the nonlinear redundant sliding surface was designed by using the nonlinear fitting function. Then, to reduce the chattering problem caused by the system, improve the input, and improve the approaching efficiency of a sliding mode surface with a large error, a nonlinear-damping, super-torsion reaching law was designed to improve the robust response efficiency of a system with a large error. For the problem of matching input interference and signal noise in the process of cable-based signal transmission, the input anomaly is non-linearly fitted by the fault-tolerant mechanism of the saturation filter, then stripped by saturation analysis, and then fitted by the nonlinear neural network. A set of nonlinear fault-tolerant subsystems was designed, which is controlled by an event-trigger mechanism. It improves the tracking accuracy of an intelligent robot in the cases of physical failure of the actuator, remote network attacks, and trajectory curvature mutation. In the comparison of simulation experiments, the pose error of this algorithm was improved by 11.48% at least. It can effectively improve the work efficiency of the freight yard and save on work costs. It has certain application prospects for

engineering. In addition, with the development of artificial intelligence research, a class of algorithms for recognition of human activities has emerged. These include semi-supervised recurrent convolutional attention model algorithms [40], adaptive semi-supervised feature analysis algorithms [41], and convolutional neural network and recurrent neural network algorithms [42]. Consider applying such algorithms to artificial intelligence transport robots. The algorithm proposed in this paper needs to be improved, and the next step will be to investigate this problem.

**Author Contributions:** Conceptualization, Y.Z. and Q.Z.; methodology, Y.Z.; software, Y.Z.; validation, Y.Z.; formal analysis, Y.Z.; investigation, Q.Z.; resources, Q.Z.; data curation, Y.L.; writing—original draft preparation, Y.Z.; writing—review and editing, Y.Z. and S.Z.; visualization, Y.H.; supervision, Q.Z.; project administration, Q.Z. All authors have read and agreed to the published version of the manuscript.

**Funding:** This research was funded by project ZR2022ME087 and the supported by Shandong Provincial Natural Science Foundation, National Natural Science Foundation of China (51911540478).

**Institutional Review Board Statement:** Not applicable.

**Informed Consent Statement:** Not applicable.

**Data Availability Statement:** Not applicable.

**Conflicts of Interest:** The authors declare no conflict of interest.

## Abbreviations

The following abbreviations are used in this manuscript:

| | |
|---|---|
| NSRSMS | New type of nonlinear, saturated, redundant sliding surface |
| NDSTRL | Nonlinear-damping, super-twisting reaching law |
| MIAC | Mean integration absolute control |
| MISE | Mean integration square error |

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
