# Peer review of "Neural Network, Nonlinear-Fitting, Sliding Mode, Event-Triggered Control under Abnormal Input for Port Artificial Intelligence Transportation Robots"

_jmse, doi:10.3390/jmse11030659_

Round 1

Reviewer 1 Report

The authors are reported a new control algorithm to solve the problems of actuator physical failure, emote network attack and sudden change of trajectory curvature with port ART tracks the transportation track in the freight yard. The nonlinear redundant saturated sliding surface is designed based on the redundant information of sliding mode control caused by the finite characteristic of control performance, and the dynamic acceleration characteristic of super-twisted sliding mode reaching law is considered to optimize the control high frequency change caused by trajectory mutation, and improved super-twist reaching law is designed. A nonlinear factor is designed to construct a nonlinear fault-tolerant filtering mechanism to compensate for the abnormal part of the unknown input that cannot be executed by adaptive neural network reconstruction. On this basis, the finite time technology and parameter event-triggered mechanism are combined to reduce the dependence on communication resources. The work is quite interesting but there is potential to further update the work, some of them are reported below,

1.      Neural network structure to be discussed through diagram, ‘https://doi.org/10.1016/j.resconrec.2018.02.025’.

2.      Training about neural network should be discussed, ‘ https://doi.org/10.1177/0959651820973797’.

3.      Key result should be discussed at the end of the abstract.

4.      Reaching time should be represented in the manuscript, 10.1177/09596518231154632 ,‘https://doi.org/10.1016/j.egyr.2022.03.159

5.      Conclusion should be rewritten to highlight the major findings.

6.      ‘’Future scope of work and drawback of the proposed work should be included.

7.       

Reviewer 2 Report

The authors of this article present a control algorithm that uses sliding mode control techniques and neural networks to control a second-order mechanical system.

It is difficult to follow the article in general due to following fundamental problems:

1.- There are a number of expressions that appear incorrect and are difficult to interpret. Several times, the authors discuss aspects that have not been introduced in a rather cryptic way, which makes reading difficult. Among these errors are the following:

a) "A new control algorithm is designed to solve the problems of actuator physical failure,

remote network attack and sudden change of trajectory curvature when port ART ..."

What is the definition of ART? Whenever an acronym is used for the first time, it must be defined.

b) "... replaces traditional transportation equipment with artificial intelligence transportation robots (ART), improves the automation ..."

This expression should be rewritten:

... replacing traditional transportation equipment with artificial intelligence transportation robots (ART), improving the automation ...

c) "It is worth noting that how to effectively reduce the buffeting effect of the algorithm itself has become a main premise of practical engineering application"

What is buffering? Maybe the auhotrs refers to chattering? Buffering is a memory operator that has no sense in this study.

d) "Using the function of eliminating redundant information of residual error by integral term, the integral saturation mechanism is designed" What is redundant information of residual error?

"The control method overcomes the shortcomings of the traditional super-twist reaching law, such as the complex adjustment of the parameters and the tendency to increase the buffeting when approaching the instantaneous mutation." What are mutations? Did the authors means the "sliding mode condition"? It is very difficult to follow the meaning of the article with this expressions.

The innovations presented by the authors should be totally rewrriten and clearly explained.

2.- Another issue with the article is the need to rewrite the equations and present them clearly and coherently.

a) The equations on page 4 must be presented correctly, indicating which variables are used and what they refer to. Lambda - what is it?

b) In Lemma 1, what is the value of e? Is this an error in the equations above? If it is a generic variable, this lemma should be introduced at the beginning of the paper or as an appendix to clarify the reading. In the event that it is an error, why is it called later qe? Additionally, references to theoretical proofs of the presented lemmas should be provided.

3.- Certain technical aspects do not appear to be addressed in the article. According to the authors, on page 2, a finite time algorithm will be introduced. The control, however, is said to be susceptible to saturation. As a result, if the workspace region is not limited, a saturated control cannot achieve finite stability time from any position in the workspace. As a result, the application region of the algorithm must be defined.

4.- The results presented should be completely rewritten, and the comments in the figures should be more descriptive.

5.- The algorithm should be compared with another state of the art solution. The comparison of results with a classical algorithm does not take into account the advances that have been suggested by other authors since the development of sliding mode control algorithms began.

Reviewer 3 Report

Check that the abstract provides an accurate synopsis of the paper. It is very vague in its present form.

The methodology of the proposed model must be illustrated by a clear flowchart.

Besides, the writing of the paper, including contributions and methodologies, should be clearer and highlight the innovation of methods & principles.

Insufficient literature is presented to support the aim of the study. This point still needs further revision.

Hyperparameters of the designed network must be included in a tabular form.

Was the algorithm trained using standard hyperparameters, or were they altered?

Comment on computational time and complexity in the training of the algorithm.

The manuscript is more like a report than a research paper failing in solid discussion. Revise results and discussion part by critically examining results and including inferences drawn. 

Reviewer 4 Report

Comments to the Author

The manuscript entitled "Neural network nonlinear-fitting sliding mode event-triggered control under abnormal input for port artificial intelligence transportation robots" has been investigated in detail. The topic addressed in the manuscript is potentially interesting and the manuscript contains some practical meanings, however, there are some issues which should be addressed by the authors:  

1.    Highlights should be added to the manuscript.

2.    The abstract should be revised and the quantitative results should be mentioned.

3.    The results are mostly expressed qualitatively, be sure to express the results quantitatively.

4.    The authors need to emphasize their contributions/novelties in the revision. In the current version, the authors did not discuss their contributions in detail.

5.    The authors should carefully proofread this paper and correct all the typos in the revision. In the current version, there are still some typos/grammar errors.

6.    The proposed algorithm still can be improved if the ideas in the following papers are explored, i.e., "Deep learning-based face detection and recognition on drones", "Making Sense of Spatio-Temporal Preserving Representations for EEG-Based Human Intention Recognition", "An Adaptive Semisupervised Feature Analysis for Video Semantic Recognition", and "A Semisupervised Recurrent Convolutional Attention Model for Human Activity Recognition". The authors are encouraged to discuss them in the revision.

7.    The references are old, use newer references.

Round 2

Reviewer 1 Report

The paper can be accepted now.

Reviewer 3 Report

The authors have addressed all my comments. Congratulations.